# Identification of transporters essential for survival of *Leishmania* promastigotes in the digestive tract of sand flies

Jovana Sádlová[1○], Ulrich Dobramysl[2,3○], Barbora Bečvářová[1], Tomáš Bečvář[1], Çağla Alagöz[4,5], Sandro Möri[4], Richard J. Wheeler[2,3], Petr Volf[1], Eva Gluenz[4,6*], Andreia Albuquerque-Wendt[1,4,6,7,8*]

**1** Department of Parasitology, Faculty of Science, Charles University, Prague, Czech Republic, **2** Medawar Building for Pathogen Research, Nuffield Department of Medicine, University of Oxford, Oxford, United Kingdom, **3** Institute of Immunology & Infection Research, University of Edinburgh, Ashworth Laboratories, Edinburgh, United Kingdom, **4** Institute of Cell Biology, University of Bern, Bern, Switzerland, **5** Graduate School for Cellular and Biomedical Sciences (GCB), University of Bern, Bern, Switzerland, **6** School of Infection and Immunity, University of Glasgow, Sir Graeme Davies Building, Glasgow, United Kingdom, **7** Parasite Chemotherapy Unit, Swiss Tropical and Public Health Institute, Allschwil, Switzerland, **8** Global Health and Tropical Medicine, Instituto de Higiene e Medicina Tropical, Universidade Nova de Lisboa, Lisbon, Portugal

○ These authors contributed equally.
* eva.gluenz@unibe.ch (EG); andreia.albuquerquewendt@swisstph.ch (AA-W)

**Editor:** Álvaro Acosta-Serrano, University of Notre Dame, UNITED STATES OF AMERICA

## Abstract

*Leishmania* amastigotes ingested by female phlebotomine sand flies are exposed to a harsh and dynamic environment that differs markedly from the intracellular niche in the mammalian host in temperature, pH and nutrient availability. Membrane transporter proteins, channels and pumps play a crucial role in maintaining cellular physiology under changing environments. A systematic loss-of-function screen of the *L. mexicana* transporter deletion mutants in macrophage and mouse infections previously identified transporter genes important for the amastigote stage. To test which transporters are important for the promastigote stage in the insect vector, we measured the fitness of gene deletion mutants in *Lu. longipalpis* sand flies. Pooled libraries of different complexities, consisting of 71–317 barcoded parasite lines allowed for an estimation of the bottleneck size in experimental infections, providing a foundation for similar experimental bar-seq studies. The fitness of each mutant parasite line was measured by tracking population composition over a course of 9 days in the sand flies and compared with the growth fitness of promastigotes over 7 days in laboratory cultures. There was a high correlation of fitness scores *in vitro* and *in vivo*, but 34 mutants showed a loss of fitness only *in vivo*, including deletion mutants of vacuolar H+ATPase (V-ATPase) subunits. V-ATPase deletion mutants expressed low levels of the metacyclic-specific transcript *sherp in vitro* and failed to generate metacyclic promastigotes in sand flies, indicating that V-ATPase function is required for parasite differentiation and progression through the *Leishmania* life cycle.

**Data availability statement:** All data supporting the findings of this study are available within the article and its supplementary materials, which have been deposited to Figshare (https://figshare.com/) under Doi: 10.6084/m9.figshare.29481254. Raw sequencing data can be accessed via ENA (https://www.ebi.ac.uk/ena) under accession number PRJEB90861 (or ERP173867).

**Funding:** AAW was the recipient of a Marie Skłodowska-Curie Individual Fellowship (trans-LEISHion-EU FP7, No. 798736) and is supported by a Marie Skłodowska-Curie Global Fellowship (LeishBlock-Horizon, No. 101148623). RJW is supported by a Wellcome Trust Henry Dale Fellowship (211075/Z/18/Z). This work was supported by a UKRI Medical Research Council grant (MR/V000446/1; This UK funded award was part of the EDCTP2 programme supported by the European Union), the Wellcome Trust (221944/A/20/Z) and the Wellcome Centre for Integrative Parasitology (WCIP) core Wellcome Centre Award (104111/Z/14/Z) and a project grant from the Swiss National Science Foundation (310030_220011). This work was partially supported by national funds through FCT – Fundação para a Ciência a e Tecnologia, I.P., under the R&D unit Global Health and Tropical Medicine (UID/04413/2025) and the Associated Laboratory in Translation and Innovation Towards Global Health REAL (LA/P/0117/2020). The funders had no role in study design, data collection and analysis, decision to publish, or preparation of the manuscript.

**Competing interests:** The authors have declared that no competing interests exist.

## Author summary

*Leishmania* parasites cause leishmaniases - a group of neglected tropical diseases that affect millions of people worldwide. These parasites must survive in two radically different environments: inside a mammalian host and within the gut of a blood-feeding sand fly. To thrive in the sand fly, *Leishmania* undergo extensive physiological changes and depend on transporter proteins to move nutrients and other molecules across their cell membranes. In this study, we focused on identifying which of these transporters are critical for the parasite's survival inside the sand fly. We used a library of genetically engineered *Leishmania* promastigotes - the parasite form adapted to the insect vector - to assess the importance of more than 300 different transporter genes. We discovered that 34 of these transporters are essential for successful colonization of the sand fly. Among them, one key protein complex - the vacuolar $H^+$ ATPase (V-ATPase) pump – was found to be crucial for parasite survival in the insect vector. Our findings deepen our understanding of how *Leishmania* adapts to life within the sand fly and highlight potential molecular targets for disrupting its transmission.

## Introduction

Protozoan parasites of the genus *Leishmania* (Kinetoplastida: Trypanosomatidae) are the causative agents of leishmaniases [1], a group of neglected tropical diseases. Over 20 *Leishmania* species share a digenetic life cycle, alternating between an insect vector and a mammalian host [1]. Females of more than 90 species of phlebotomine sand flies - *Phlebotomus* in the Eastern Hemisphere and *Lutzomyia* in the Western Hemisphere - serve as their primary vectors [2].

When a female sand fly takes a blood meal from an infected mammalian host, it ingests immotile amastigote forms, which are encased in a chitin-rich peritrophic matrix (PM) [2]. During this stage, parasites face dramatic environmental changes, including a temperature drop (from ~37 °C to ~26 °C), pH shift (from acidic to neutral/alkaline), and altered nutrient availability (from blood components to sugar meals and microbiome metabolites). These signals trigger rapid differentiation from amastigote into promastigotes, often within 6 hours post feeding [3,4].

Inside the sand fly gut, *Leishmania* promastigotes differentiate into a series of different morphotypes [4]. The initial amastigote to procyclic promastigote differentiation is marked by key metabolic changes, including a ~10-fold increase in uptake of carbon sources (e.g., glucose and non-essential amino acids), and elevated secretion of glycolytic end-products [5–8]. These weakly motile, short-flagellated procyclic forms undergo binary fission for at least 48–96 hours before slowing down their replication and differentiating into highly motile elongated nectomonad promastigote forms [9]. After 72–96 hours post blood feeding, the toxic products of blood digestion cause a reduction in the number of procyclic forms [10] and enzyme-mediated PM disintegration ensures that nectomonads escape the PM-encased blood meal into the

midgut lumen [9,11]. There, they bind to the midgut epithelium via parasite- and vector-derived surface molecules such as lipophosphoglycans (LPGs) or mucin-like O-glycoconjugates, which prevent their expulsion during defecation [12–15]. Parasites of the *Leishmania* subgenus then differentiate into replicative leptomonad promastigotes and migrate to the anterior midgut [3]. During this phase, flagellar motility is crucial for *Leishmania* parasites to migrate through the thoracic midgut toward the stomodeal valve, the structure at the junction between the sand fly foregut and midgut – a process required for transmission. For example, Beneke et al. (2019) used genetically engineered *L. mexicana* promastigotes with impaired flagellar motility to infect *Lu. longipalpis* and observed that these species require directional motility to successfully colonise the fly [16]. Furthermore, Cuvillier et al. (2003) showed that overexpression of constitutively active variant of the ADP-ribosylation factor-like protein 3A (ARL-3A) in *L. amazonensis* promastigotes resulted in cells with a short, non-motile flagellum, which failed to colonise *Lu. longipalpis* sand flies [17]. When reaching the stomodeal valve, parasites undergo differentiation into two main morphologically distinct forms: replicative, short-flagellated, non-motile haptomonads and non-replicative, long-flagellated highly motile metacyclic promastigotes [18,19]. Notably, recent single- cell transcriptomic evidence from *L. major* promastigotes isolated from *Ph. duboscqi*, suggest that metacyclic promastigotes may be further divided into two transcriptionally distinct sub-forms; replicative early metacyclics and non-replicative late metacyclics [4]. Moreover, the same study provides convincing evidence that in addition to metacyclics, which are traditionally viewed as the primary infective stage, haptomonads also play a significant role in transmission [4].

In mature sand fly infections, parasites secrete promastigote secretory gel (PSG) - a viscous matrix rich in filamentous proteophosphoglycans (fPPGs) - which fills the thoracic midgut [20–24]. In addition, they secrete chitinase [25], which damages the insect´s alimentary canal [26]. Along with fPPGs, this disruption alters sand fly feeding behavior and promotes regurgitation during subsequent blood meals [12,27,28]. Collectively, these changes enhance parasite transmission by increasing the number of parasites egested into the skin of the mammalian host [28].

Although our understanding of the *Leishmania* life cycle within sand flies still lags behind that of other vector-borne diseases, several molecular determinants of vector colonisation have been identified [3,12,15,16,24,29–36,37–41]. Recent advances in reverse genetics and high-throughput barcode sequencing (bar-seq) in *Leishmania* sp. [42] have significantly accelerated discovery of parasite genes essential for promastigote fitness and vector colonisation [16,32,43]. For instance, proteins required for flagellar assembly (IFT88, LmxM.27.1130) and motility (e.g., the central pair protein PF16, LmxM.20.1400 or the inner dynein arm protein IC140, LmxM.27.1630) [16] were found to play a crucial role in persistence in sand flies and migration to the stomodeal valve [17]. In a separate screen targeting the parasite's kinome, ATM (LmxM.02.0120) and PI4K (LmxM.33.3590), were identified as atypical (aPK) and phosphatidylinositol 3' kinase-related (PIKK) protein kinases, respectively, conditionally essential for survival only in sand flies, suggesting unique pathways are involved in vector-stage survival [32]. Another kinase, MPK9 (LmxM.19.0180), was identified as essential for sand fly colonisation [32]. In an independent study, MPK9 was shown to influence flagellar length [44], reinforcing the importance of flagellum integrity for the parasite survival inside the vector [44]. Moreover, single-cell RNA sequencing is beginning to reveal molecular markers of parasite development in insect stages [4,45]. Despite these advances, the role of transporter proteins in sand fly colonisation remains largely underexplored. Exceptions include two nucleotide sugar transporters involved in lipophosphoglycan (LPG) biosynthesis. One, LPG2 (LmxM.33.3120), encodes a GDP-mannose transporter responsible for incorporating the initial and repeating mannose units into the mannose-rich LPG structure and was shown to be essential for development of *L. donovani* and *L. major* in several sand fly species, including *Ph. argentipes*, *Ph. papatasi*, *Ph. duboscqi* and *Ph. perniciosus* [31,33,35,46]. Another deletion, that of the LPG5A/B gene array (LmxM.24.0360-65) encoding a UDP-galactose transporter, resulted in reduced colonisation of *L. major* in *Ph. duboscqi* [35]. This transporter incorporates galactose units into the same repetitive LPG backbone, highlighting the importance of glycan modifications in vector attachment and colonisation. This is further supported by ablation of additional (non-transporter) genes involved in LPG biosynthesis [15,16,31,46]. Interestingly, LPG mediated binding between sand flies and *Leishmania* is known to be crucial in parasite-vector systems where the vector is specific, i.e., each glycoconjugate

presented by each different *Leishmania* sp. enables extraordinary specificity to a single vector species [47,48]. In contrast, permissive vectors such as those within the genus *Lutzomyia*, have been reported to present additional binding mechanisms involving O-glycosylated proteins [49].

Recently, we systematically assessed the fitness contribution of 312 predicted *L. mexicana* membrane transporters, channels and pumps during the exponential phase of promastigote growth *in vitro*, in amastigotes in human induced pluripotent stem derived macrophages (iMACs) and over 6 weeks in mice (*in vivo*) [43]. Using bar-seq, we showed that deletion of at least 40 transporters compromised amastigote survival *in vivo* [43]. The vacuolar H$^+$ ATPase (V-ATPase) emerged as a crucial proton pump for the survival of parasites *in vivo*, and *in vitro* under conditions of low external pH [43]. Here, we extend that work by conducting an independent comprehensive systematic loss-of-function screen targeting 316 single putative transporter-encoding genes and 17 gene arrays. We assessed mutant fitness over a one-week time course *in vitro* and in a sand fly model of infection *in vivo*. This screen revealed some mutants that show gain-of-fitness phenotypes and many with loss-of-fitness phenotypes. While there was a positive correlation between mutant fitness *in vitro* and in the flies, these results indicate a vital function for ion pumps, sugar nucleotide transporters, and transporters of some other classes, notably several mitochondrial carrier proteins, for survival and fitness within their sand fly vector. Moreover, the V-ATPase is required for effective completion of the developmental cycle from promastigotes to metacyclics *in vivo*.

## Results and discussion

### An expanded gene deletion screen of the *L. mexicana* transportome reveals that most transporters are dispensable for promastigote survival *in vitro*

To interrogate the fitness phenotypes of mutant promastigotes over an extended time course in laboratory cultures and in experimental infections of *Lutzomyia longipalpis* sand flies, we generated new CRISPR-Cas9 mutant libraries to allow for the inclusion of additional genes and gene arrays, and to assemble pooled libraries of different complexities for the sand fly infections. In addition to the previously defined "transportome" [43] four newly identified genes were targeted: two from the Acetate Uptake Transporter (AceTr) family, one with similarity to the Selenoprotein P Receptor (SelP-Receptor) family, and one Multidrug/Oligosaccharidyl-lipid/Polysaccharide (MOP) Flippase, as classified by the Transporter Classification Database (TCDB) [50], thus expanding the number of proteins in the *L. mexicana* "transportome" to 316 (S1 Table) [16,42,43,51]. This approach successfully generated 304 viable mutant populations, each resistant to both blasticidin and puromycin selection drugs (S1 and S2 Tables), which were broadly organised by TCDB families into four sub-libraries (S4 Table). Diagnostic PCR confirmed the successful deletion of all copies of the targeted genes for 154 lines (null mutants); the remaining 132 lines where the targeted gene was still detectable were classified as 'refractory to deletion' (S3 Table). Of the confirmed single-gene deletions, 46 had not been successfully generated in our previous screen [43]. This new mutant library also identified three additional single-member superfamilies whose transporters appear dispensable *in vitro*, namely the mitochondrial EF hand Ca$^{2+}$ uniporter regulator (MICU; *LmxM.07.0110*), the Proton-dependent Oligopeptide Transporter (POT; *LmxM.32.0710*) and the Selenoprotein P Receptor (SelP-Receptor; *LmxM.28.2380*) in addition to the eight superfamilies that were previously shown to be dispensable [43].

### Deletion of genes arranged in tandem arrays

A total of 17 gene arrays were targeted, including nine not previously targeted, from the following families: AceTr (1 array), Amino Acid/Auxin Permease (AAAP, 4 arrays), Cyclin M Mg$^{2+}$ Exporter (CNNM, 1 array), Equilibrative Nucleoside Transporter (ENT, 1 array), Major Facilitator (MFS, 2 arrays), Mitochondrial Carrier (MC, 4 arrays), P-type ATPase (P-ATPase, 1 array), Voltage-gated Ion Channel (VIC, 2 arrays), Zinc (Zn$^{2+}$)-Iron (Fe$^{2+}$) Permease (ZIP, 1 array) (S1 Table). Upon analysis of the 13 drug resistant mutants that survived the selection, only one AAAP array mutant (*LmxM.34.5350* and

*LmxM.34.5360*) was found to be null (S1 and S3 Tables). For the *LmxM.18.1290* and *LmxM.18.1300* array where a null mutant was previously achieved [43], only double puromycin, blasticidin resistant mutant populations retaining at least one copy of the targeted gene were recovered (S1 and S3 Tables and S3 Fig). As discussed previously [43], technical challenges such as high sequence similarity and underestimated gene copy numbers can hinder the isolation of null mutants from tandem arrays. For instance, at first, we failed in the isolation of a null mutant for the glucose transporter array which harbours three genes (LmGT1-LmGT3), despite its successful deletion in *L. mexicana* using a strategy of gene knock-out by homologous recombination [52,53]. In this case, we repeated transfections using both puromycin and blasticidin selection, but only double drug-resistant populations emerged that had retained at least some LmGT genes. It was only after a third round of transfection and subsequent selection of clonal cell lines that we successfully isolated three clones lacking the entire array (S6A - S6C Fig). The doubling times of two null mutant clones were significantly increased (8.73 and 8.24 h) compared to that of the parental cells (5.14 h) (S6D – S6E Fig). This suggests that although the GT-array null mutants are viable, mutants that somehow retained one or several of the genes from the targeted array may have a significant growth advantage in mixed populations. These data show that while it is possible to achieve array null mutants, the technical challenges in identifying and isolating array mutants precludes phenotype screens at scale with this bar-seq method.

### Less than 30% of the *Leishmania* transportome is essential for promastigote survival

Consolidating data across the two independently generated libraries indicates that 225 (~71%) transporter-encoding genes are dispensable for promastigote survival in standard *in vitro* laboratory cultures (S3 Table and S1 and S2 Figs) [43]. The successful deletion of these genes is positive proof that they are not essential for cell proliferation under the tested conditions, although they may still contribute to fitness. Conclusive statements cannot be made however about the importance of genes where a deletion attempt failed. For the 91 genes refractory to deletion in two independent screens (S3 Table), further attempts at gene deletion may yet prove successful. Data released from the LeishGEM genome wide gene deletion screen (https://browse.leishgem.org/) [54] already reports several transporter gene deletion mutants that were not retrieved in the current screen. Examples include transporter-encoding genes located on supernumerary chromosomes like the putative acidocalcisome inositol 1,4,5-triphosphate receptor/$Ca^{2+}$ release channel (LmxM.16.0280), one of the four amino acid transporters of the AAT1 locus, AAT1.4 (LmxM.30.0350), a glycosomal ABC transporter, GAT1 (LmxM.30.0540) and a major facilitator (LmxM.30.0720); on diploid chromosomes a porphyrin transporter (LmxM.17.1430), the amino acid transporter, AAT23.2 (LmxM.27.0680) and a mitochondrial carrier (LmxM.29.2240) (S3 Table). Conditional gene knockout strategies would be required for conclusive functional validation and stronger support for a claim of "essentiality" [55].

### Prolonged culturing of transporter deletion mutants identifies novel growth fitness phenotypes *in vitro*

We next asked whether gene disruption had any effect on the relative growth rates of the surviving promastigotes in standard laboratory cultures, over a one-week time course. To assess their relative fitness, all 304 viable isolated barcoded transporter mutants and 13 array mutants, were combined into a single masterpool. To this pool we added five barcoded parental control lines (SBL1–5), eight non-transporter null mutants with previously characterised phenotypes [three independently barcoded ΔLPG1 (*LmxM.25.0010*, normal growth, important for sand fly colonisation), two independently barcoded ΔPF16 (*LmxM.20.1400*, normal growth, essential for sand fly colonisation), three independently barcoded ΔIFT88 (*LmxM.27.1130*, very slow growth, essential for sand fly colonisation) [17], and one non-transporter null mutant ΔLmxM.15.0240 (nonspecific lipid-transfer protein) [43] (S4 Table). This masterpool was split into three separate flasks and grown in standard M199 culture medium for seven days. Cultures were diluted into fresh medium twice during this period (Fig 1). DNA was sampled at baseline (0 hours), and after 24, 48, and 144 hours (Fig 1) and each mutant's relative

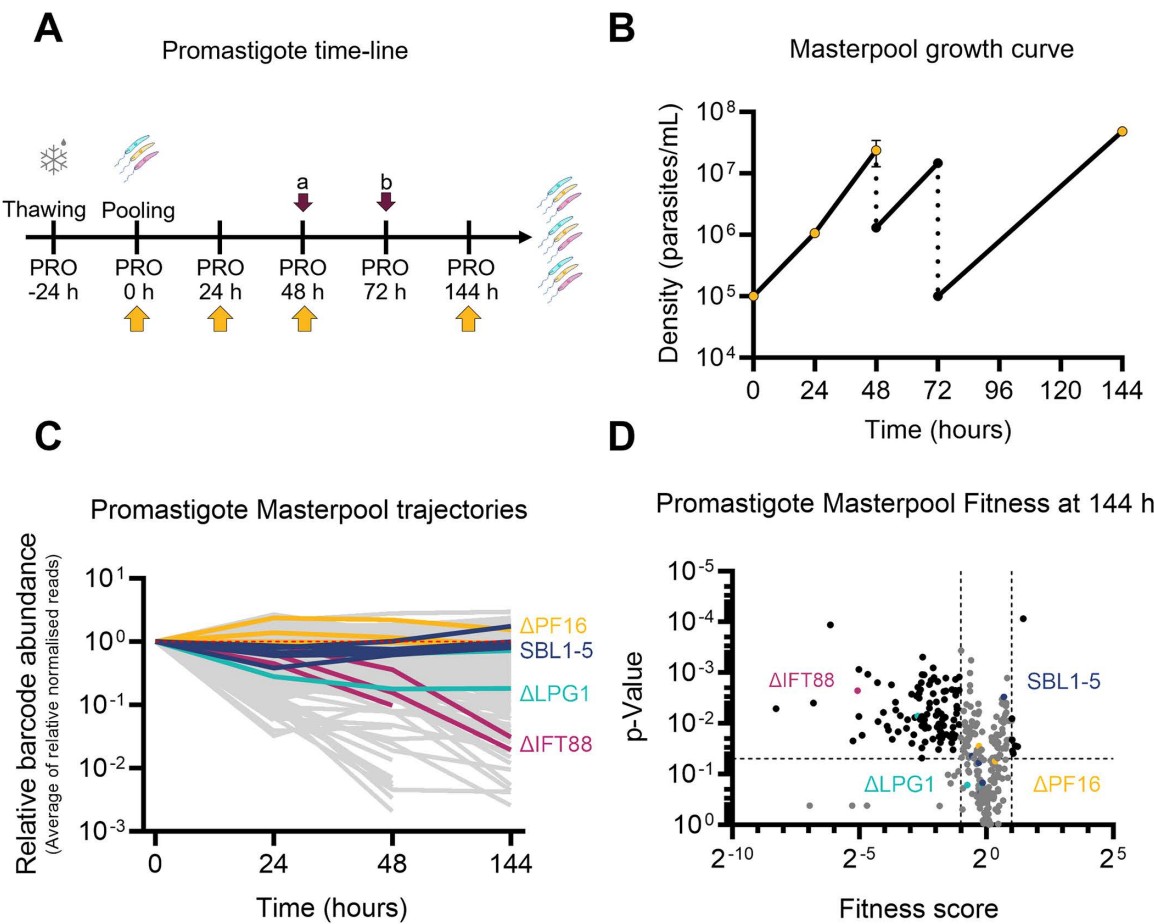

**Fig 1. Fitness of promastigote mutants *in vitro*. (A)** Overview of the experiment timeline for *in vitro* growth of promastigotes in culture. DNA sampling time-points are indicated by yellow arrows. Dark pink arrows denote dilutions, to (a) 1 x 10⁶ parasites/ml and (b) 1 x 10⁵ parasites/ml. **(B)** Growth profile of the masterpool of promastigote mutants over time. Data points are the average of three measurements; yellow dots indicate where gDNA was sampled. The dotted line indicates dilution of the cultures. **(C)** Trajectories of the average of normalised reads of the promastigote masterpool, relative to time-point "0 hours" (T₀). Red dotted line highlight relative barcode abundance of 1. Controls are shown in colour: dark blue, SBL1-5 parental cell lines; Cyan, ΔLPG1; Magenta, ΔIFT88; Yellow, ΔPF16. Grey, all other barcoded cell lines. **(D)** Volcano plot showing fitness scores against p-values of mutants from the promastigote masterpool after 144 hours of growth. Dashed lines demarcate fitness score thresholds of < 0.5 and > 2, and a significance threshold of p<0.05. Barcodes meeting both threshold criteria are coloured black, non-significant values are grey. Controls are coloured as in **D.**

representation over time was assessed by measuring DNA barcode abundance at the sampled time points and calculating the proportion of each barcode at a given time point relative to its representation in the starting population (S5 Table). This showed that the parental control cells and a majority of the mutants maintained a flat trajectory, indicating that they proliferated in the population at similar rates. There were a small number of mutants with an upwards trajectory, indicating their proportion within the pool increased over time, and a larger set where barcode proportions sharply decreased over time (Figs 1C, S4, and S5 and S5 Table). To quantify this, fitness scores were calculated by comparing the change in barcode proportions for a given mutant to that of the mode of the cell line change distribution (S5 Table). After 144 hours, seven mutants displayed enhanced fitness in the promastigote *in vitro* pool (fitness score above 2 and p<0.05, Fig 1D): two mutants lacking a mitochondrial carrier (Δ*LmxM.15.0120* and Δ*LmxM.34.3330*), two ABC transporters (Δ*LmxM.06.0100* and Δ*LmxM.11.1290*), two amino acid permeases (Δ*LmxM.30.1820* and Δ*LmxM.27.0680*) and one hypothetical protein

of the MSF family (Δ*LmxM.34.2810b*). In contrast, 107 transporter mutants exhibited significantly reduced fitness (score below 0.5 and $p < 0.05$, Fig 1D), with two barcodes dropping below the detection limit at 48 h and eleven at 144 h (zero read counts in all three replicates, S5 Table). Amongst the mutants disappearing rapidly from the population was a confirmed deletion of ABCB3, which acts in heme and cytosolic iron/sulfur clusters biogenesis and is required for *L. major* virulence [56]. This severe loss of fitness in culture may explain why previous attempts to generate null mutants for this transporter were unsuccessful [43,56]. Still detectable at the lowest levels were confirmed null mutants for a predicted sodium/hydrogen exchanger of the CPA1 family (Δ*LmxM.14.0980* score 0.014, $p = 0.0001$) and a putative calcium-transporting P-ATPase (Δ*LmxM.32.1010*, score 0.009, $p = 0.004$). Also significantly depleted were mutants for predicted ADP/ATP carrier proteins: Δ*LmxM.07.0530* (MCP15, refractory to deletion, MCP nomenclature taken from [57]) and the tandem array *LmxM.19.0200_LmxM.19.0210* (MCP5, refractory to deletion). The deletion of the fourth ADP/ATP carrier predicted in the *L. mexicana* genome (LmxM.14.0990, MCP16) resulted in a less severe but also significant loss of fitness (score 0.17 and $p = 0.0016$), indicating that these mitochondrial carrier proteins perform vital non-redundant functions in promastigotes.

In this study, the culture conditions were designed to provide ample nutrients, a buffered environment and constant temperature. However, over the 144 hours, cells experienced changes in population density and two culture dilutions. After 48 h of growth and a density of $>3 \times 10^7$ parasites/ml, the population doubling time increased from 5.36 h to 6.87 h, suggesting entry into stationary phase just before dilution (S6 Table). Consequently, these mutants experienced a variety of stresses, including microenvironmental pH shifts, nutrient depletion, and waste metabolite accumulation. Thus, although the design of the knockout libraries has the inherent limitation that only viable mutants can be assessed, these changeable conditions allowed for the detection of fitness phenotypes and indicate that many of the null mutants would be outcompeted over time in mixed populations. Future work comparing transporter requirements across different media and culture conditions may reveal additional phenotypes. While M199 medium is widely used for *Leishmania* cultivation, it was originally developed for fibroblasts [58] and is unlikely to reproduce the exact conditions encountered within the sand fly gut.

### Loss and gain of fitness in transporter deletion mutants *in vivo*

We next asked which mutants would be able to tolerate this more varied and harsher environment in the alimentary tract of phlebotomine sand flies. To evaluate the relative fitness of all viable transporter mutants *in vivo*, we created four sub-pools (named P1-P4) which were added separately to blood feeds presented to female *Lu. longipalpis* sand flies. Each pool contained five barcoded parental control lines (SBL1–5), three non-transporter null controls with established *in vitro* and *in vivo* promastigote phenotypes (ΔLPG1, ΔPF16, ΔIFT88), and on average 86 barcoded mutants per pool. We reasoned that this smaller pool size would reduce the chance of mutants being lost at random considering the small volume ingested by the sand flies. Assuming that in experimental conditions each *Lu. longipalpis* female feed 0.8-1 µl of blood [59], each fly would be expected to ingest 8´000–10´000 parasites if the blood-cell suspension was prepared at $1 \times 10^7$ parasites/ml, guaranteeing an average of 93–116 parasites per mutant line, from a pool of 86 barcoded mutant lines – a level within the range used in comparable bar-seq studies [16,32]. We also infected sand flies with the masterpool to allow for a direct test of the effect of bigger pool sizes on the outcome of the bar-seq screen. DNA was collected from the parasite-blood mixture at 0 hours (pre-infection) and from infected sand flies after 2 days (48 h) and 9 days (216 h) post blood meal (PBM) (Fig 2A). The barcode proportions for each mutant at each time point were quantified by sequencing.

### Bar-seq performance by pool size and infection bottlenecks

To confirm the validity of our pool sizes and evaluate how many cell lines go extinct due to population bottlenecks in the process of establishing an infection, we turned to a more comprehensive mathematical analysis. First, we developed an

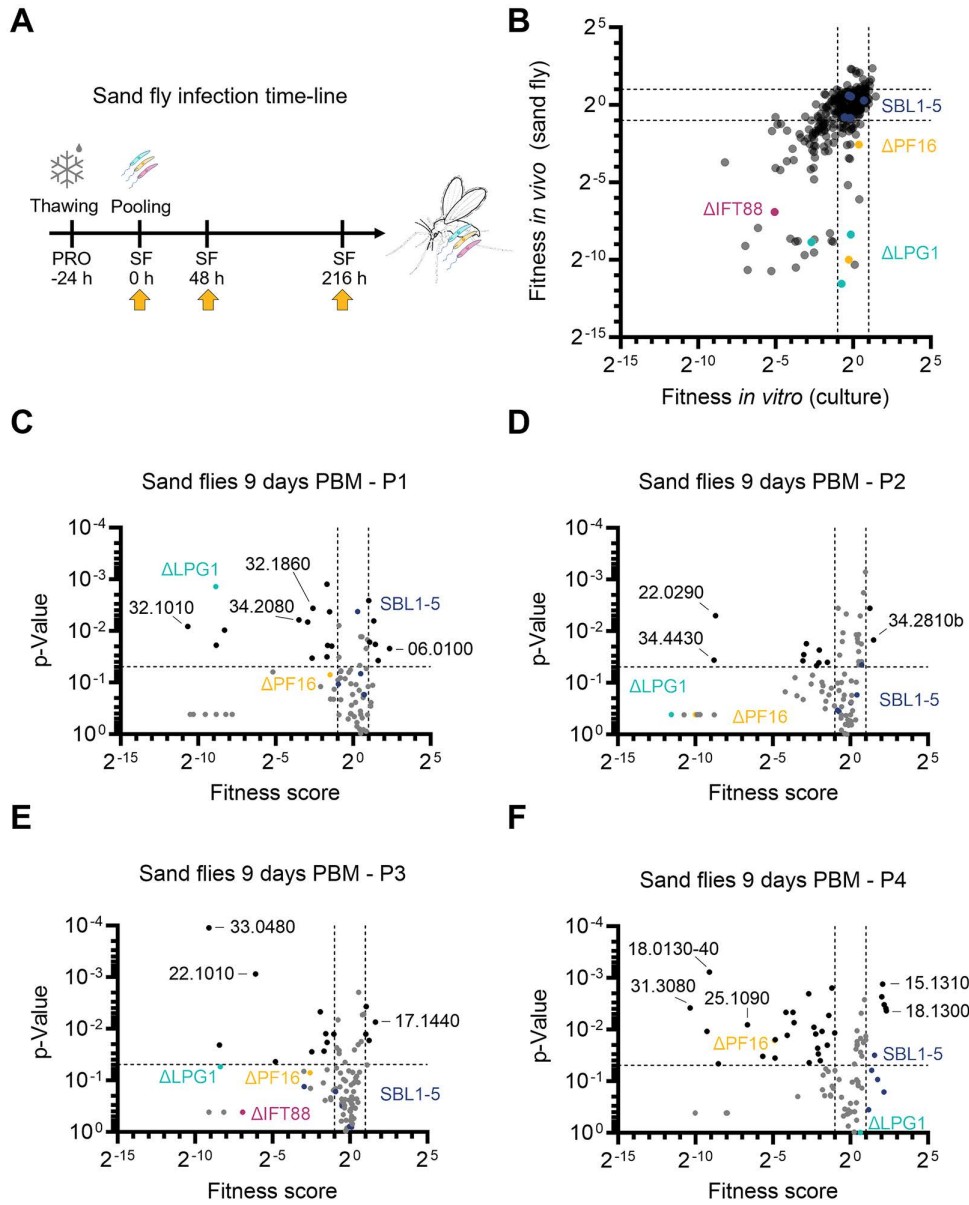

**Fig 2. Fitness of promastigote mutants *in vivo*. (A)** Overview of the experiment timeline for sand fly infections. DNA sampling time-points are indicated by yellow arrows. **(B)** Fitness scores from the promastigote masterpool after 144 hours in vitro growth plotted against the fitness scores from all mutants 216 hours after infection of sand flies. Black dashed lines mark fitness score thresholds of < 0.5 and > 2. **(C-F)** Volcano plots showing fitness scores against p-values after 216 h (9 days PBM) in sand flies, separated by sub-pools (P1-P4). Dashed lines demarcate fitness score thresholds of < 0.5 and > 2, and a significance threshold of p < 0.05. Barcodes meeting both threshold criteria are colored black, non-significant are grey. Controls are shown in colour: Dark blue dots, SBL1-5 parental cell lines; Cyan, ΔLPG1; Magenta, ΔIFT88; Yellow, ΔPF16. Black numbers denote the abbreviated GeneIDs (LmxM.xx.xxxx) of selected mutants with the highest fitness changes.

evenness score representing how uniformly the mutant lines were represented in the pools before the start of the assay. As a guide, higher values are better and a value below 0.5 indicates a pool in which a few mutant lines dominate, with other cell lines having low barcode read counts (see Materials and Methods). This evenness parameter was on average 0.79 for pool P1, 1.24 for pool P2, 1.02 for pool P3, 0.87 for pool P4, and 1.06 for the masterpool, with a standard

deviation across the pool replicates of less than 0.03 in all cases (Fig 3A). Next, we checked how many cell line barcodes became undetectable in each pool's replicates at the 2-day and 9-day time points as an estimate of how many barcodes became extinct. Using a Pólya urn model we then estimated that between 16 and 265 cells (with a median of 50 cells) establish an infection within a single sand fly (Fig 3B; see Materials and Methods for the mathematical derivation). This indicates that for pools P1-P4 the number of flies chosen in the experiment was sufficient, using the criterion that 90% of hypothetical equal-fitness cell lines survive the experimental population bottleneck on average. However, the number of flies chosen for the masterpool was not sufficient to reach this threshold, due to the high number of cell lines it contains despite good evenness (Fig 3C), therefore was not included in subsequent analyses. It is important to meet the 90% survival threshold, otherwise the interpretation of low-proportion barcodes and extinction as cell line fitness becomes difficult to disentangle from random extinction events.

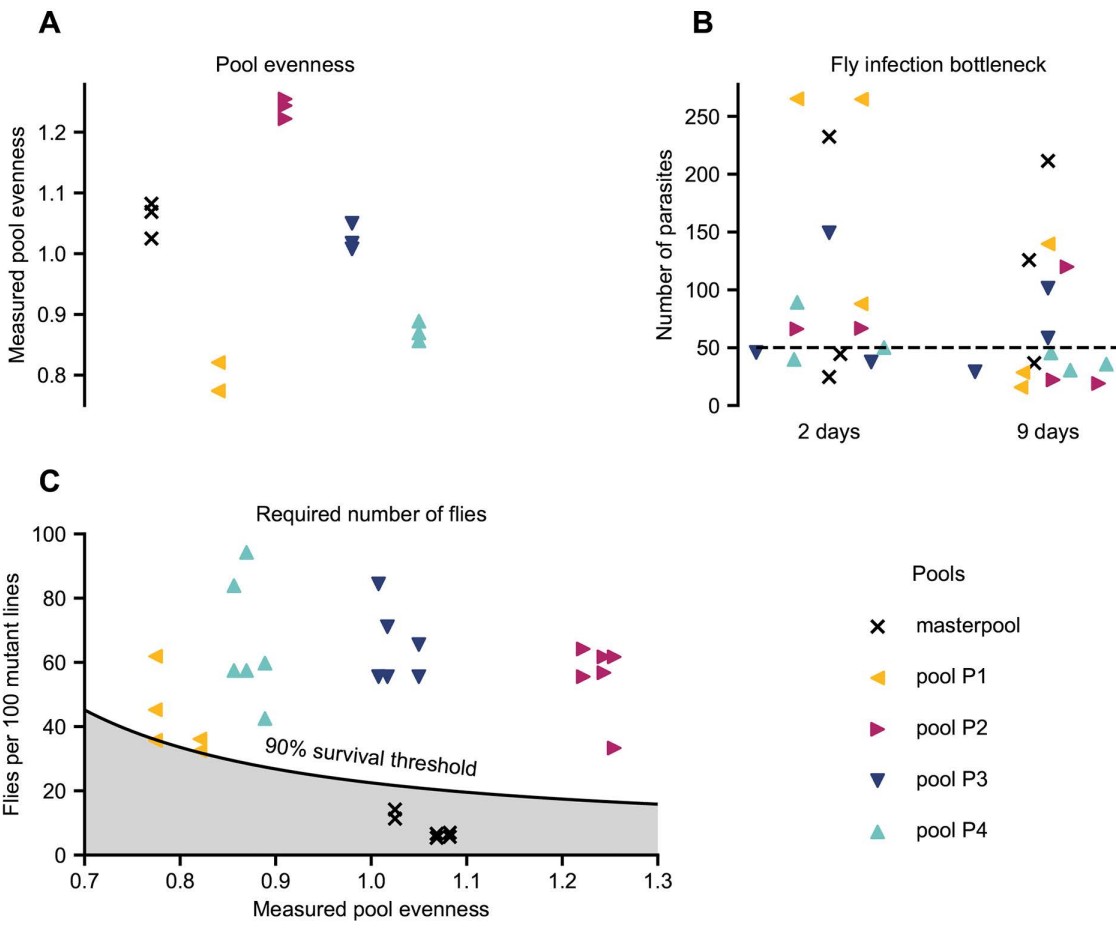

**Fig 3. Pool bottlenecks and required number of flies. (A)** Pool evenness. A small value indicates that the pool is dominated by a few barcoded cell lines with high cell counts. Conversely, a large value means that all cell lines have comparable cell counts. Normal values for pools of cell lines fall into the range 0.7-1.3. **(B)** Inferred per-fly population bottleneck sizes. Each marker shows the average number of cells that successfully established a detectable infection within a single sand fly for a given replicate, time point and pool according to our mathematical model. The dashed line shows the median population bottleneck size (50 cells per fly). **(C)** Number of flies used in each pool replicate vs. number of flies required to meet a 90% survival threshold. The solid black line shows the threshold number of flies needed as a function of the pool evenness such that 90% of cell lines survive on average if all cell lines share the same fitness. If this criterion is not met (grey shaded area), it becomes difficult to interpret low-proportion barcodes and extinction as indicators for fitness rather than random events. All pools except for the masterpool lie above the threshold, hence the masterpool was excluded from further analysis.

In each cohort, the parental control lines remained stable over the 9-day infection (S5C – S5F Fig), with many of the mutant barcodes following the same trajectory as the parentals, indicating no fitness change. The trajectories of the control mutants ΔIFT88, ΔLPG1, and ΔPF16 indicated depletion of these mutants, as expected [16], albeit with some variation between the different pools (S5C - S5F Fig). To quantify these changes, fitness scores were calculated to identify mutants that became significantly depleted or enriched in the flies (S5 Table). The fitness scores of the mutants 9 days PBM in flies showed a positive correlation (Pearson r = 0.6055) with the fitness scores of promastigotes measured after 144 h in culture (Fig 2B). Across all sub-pools, 80 barcoded transporter mutants displayed significantly reduced fitness (score < 0.5, p < 0.05), while nine mutants showed increased fitness (score > 2, p < 0.05) in the sand fly (Fig 2C - 2F).

There was a small number of mutants that showed a loss-of-fitness phenotype only in the fly (Fig 2B), including a UDP-Gal nucleotide sugar transporter (LmxM.22.1010 HUT1L), one amino acid transporter (LmxM.32.1420), two ABC transporters (LmxM.32.3260 ABCI4 and LmxM.32.1800 ABCD3, the ortholog of the *T. brucei* glycosomal transporter GAT2 [60]), one uncharacterised MSF transporter (LmxM.01.0440), a putative calcium motive p-type ATPase (LmxM.34.2080) and a subunit of the V-ATPase (discussed below). While the substrates of most of these transporters remain to be defined, they may contribute to the parasite's metabolic adaptation to changing environments in the fly, or to the secretion of glycoconjugates (e.g. HUT1L). The importance of glycoconjugate secretion *in vivo* is supported by the phenotype of the control mutant ΔLPG1, which became strongly depleted in the flies. Loss of the Golgi GDP-Man transporter LPG2, which lacks a broader range of glycoconjugates including LPG [33,35,46,61], also resulted in mutants with low fitness scores in flies [16], as well as in *in vitro* cultures, although these did not pass the statistical significance test.

The gain-of-fitness mutants included three ABC transporters; ABCG3 (LmxM.06.0100), ABCH1 (LmxM.11.0040) and ABCA6 (LmxM.11.1290), one MFS protein (LmxM.34.2810), one UDP-galactose transporter (LmxM.24.0365), aquaglyceroporin 1 (AQP1, LmxM.30.0020), two folate-biopterin (FBT) transporters; FT1 (LmxM.10.0400) and LmxM.19.0920 and one voltage-gated calcium channels (VGCC) of the VIC family (LmxM.17.1440). The latter encodes one of two L-type VGCCs in *Leishmania*, previously shown to be sensitive to VGCC inhibitors [62]. Interestingly, ΔLmxM.17.1440 promastigotes also exhibited significantly increased fitness *in vitro* after 24h and 144 h (this study and [62]) while their abundance decreased in macrophages at 120 h (score = 0.5, p < 0.05 [62]). In contrast, the second VGCC (LmxM.33.0480) showed consistently reduced fitness both *in vitro* and *in vivo*, suggesting that these VGCCs have distinct roles in calcium homeostasis across life cycle stages. Similarly, stage-specific phenotypes were observed for the two predicted magnesium transporters of *L. mexicana*. Mutants where MGT2 (LmxM.25.1090) was targeted, but only double drug-resistant populations, refractory to gene deletion were isolated, resulted in decreased fitness in sand flies (score 0.01, p = 0.008), while the MGT1 mutant (ΔLmxM.15.1310, null) was enriched in the flies 9 days PBM. An MGT1 null mutant was previously found to have significantly reduced fitness in mice [43]. Prior work suggests AQP1 may function in volume regulation, osmotaxis and antimony [Sb(III)] uptake [63], although its broader biological role remains unclear. While direct measurements sand fly gut or macrophage phagolysosome osmolality are unavailable, estimates suggest values range from 85-448 mOsm/kg in a hematophagous insect gut [64] and 275–295 mOsm/kg in mammalian blood [65]. The phagolysosome is likely mildly hyperosmotic due to ion influx, digestion and acidification. How these osmotic shifts across host environments are buffered in the absence of AQP1 remains to be investigated. Similarly, the phenotypes of the FBT family mutants requires further study. Folate transporter FT1 (LmxM.10.0400) is known to be highly expressed in actively dividing promastigotes [65–67]. *Leishmania*, like many other eukaryotic cells, are folate (vitamin B9) auxotrophs, requiring external folate uptake. Host folate availability can vary widely depending on nutritional status and microbiota, implying that *Leishmania*'s 13 FBTs (S1B Fig) may be specialised for stage-specific roles. The mutant for the FBT family biopterin transporter BT1 (ΔLmxM.34.5150) was significantly depleted from flies 9 days PBM (score 0.003, p = 0.02) and completely lost from the promastigote *in vitro* culture.

Whether the higher fitness scores reflected more rapid proliferation or better survival or persistence in the fly following excretion of the digested bloodmeal cannot be deduced from these data. Similarly, a loss of barcodes from the population

could indicate a higher death rate or slower proliferation. In the *in vitro* assay, exponential growth of the population was precisely measured, showing a rate of 20.3 doublings over the 144 h time course (S6 Table). In the fly, the promastigotes normally progress through a series of replicative and non-replicative developmental stages. How many exact doublings a wild type *L. mexicana* promastigote is expected to undergo during 9 days in *Lu. longipalpis* is not precisely determined, but it is likely to be fewer than in the constant environment of a culture flask [12]. Previous reports, suggest that promastigotes may take ~29 h (S6 Table) to double in the digestive tract of a female sand fly [12]. However, this is a very rough estimate, since factors like parasite death or loss from the fly during defecation of bloodmeal remnants, are likely the dominant reasons for severe dropouts in the sand fly. Thus, small differences in growth rate would be hard to detect in the fly, but easily detectable in culture, especially in the high-glucose M199 medium that maximises the growth rate of *Leishmania* promastigotes. Likely as a result of that, we observed a greater number of mutants with small but significant changes in fitness *in vitro* (107/304) compared to *in vivo* in sand flies (80/304) (Fig 2B and S5 Table). We can speculate that this reflects differences between the assays, where the *in vitro* populations underwent a larger number of doublings, combined with the technical advantage of sampling larger amounts of DNA from the *in vitro* cultured cells, allowing for the reliable detection of small differences in replication rate, which may not be detectable in the fly assay.

Future experiments, assessing fitness of different mutants and morphotypes within defined sections of the sand fly gut, and in flies may provide further insights into the functions of specific transporter proteins, as well as exposing sand flies to multiple blood meals, which can alter promastigote differentiation and vector competence [68].

## The V-ATPase of *L. mexicana* is critical for survival and parasite differentiation in the sand fly gut

We recently showed that loss of V-ATPase subunits had little effect on the growth of promastigotes in standard culture medium at neutral pH but it proved detrimental to promastigotes in acidified culture medium, and caused significant loss of fitness in macrophages and mice [43]. Here, we found that loss of V-ATPase subunits also had a profoundly detrimental effect on promastigotes *in vivo*. Consistent with the finding that V-ATPase subunits are non-essential for promastigotes in culture, the barcode trajectories of V-ATPase mutants showed only a moderate decline after 48 hours of promastigote *in vitro* growth (Fig 4A) when the cells had reached a density of >1 x $10^7$ cells ml$^{-1}$ (Fig 1B); at that density, dilution into fresh medium was required to maintain the population in log phase. Despite the decline in abundance, the barcodes of V-ATPase mutants were still represented within the pool after 144 h of continued exponential growth. In the sand flies, the decline was more pronounced within just 48 h PBM (Fig 4B), when the infected bloodmeal was still surrounded by peritrophic matrix, and declined further at 9 days PBM, resulting in lower fitness scores than observed *in vitro* (S5 Table).

To investigate this phenotype further, we introduced an ectopic copy of the V-ATPase subunit E (V$_1$E, Fig 4C) into the V$_1$E null mutant cell line ($\Delta LmxM.36$.3100, KO) to generate an addback control (AB) and compared the growth profiles of the three cell lines *in vitro* and *in vivo* (Fig 4D). The growth rates of the KO, AB and parental cell lines were measured over 6 days, with one dilution at the end of log phase on day 3. All three cell lines grew at a comparable rate for the first three days; after dilution the null mutant cell line continued exponential growth but with a slightly longer doubling time. The growth of the AB cell line was indistinguishable from that of the parental control (Fig 4D). We next infected female *Lu. longipalpis* flies with the KO, AB or the parental *L. mex* Cas9 T7 cell line to measure parasite abundance, location, and developmental stages over time (Fig 4E - 4G). At 48 h PBM, 100% of flies infected with parental and AB lines showed heavy infections (> 1000 promastigotes/gut, Fig 4E) and parasites were located within the endoperitrophic space surrounded by the peritrophic matrix (Fig 4F). In contrast, the KO persisted in only 40% of flies, and infections were mostly moderate (100–1000 promastigotes/gut). By day 9 PBM, most flies infected with parental and AB lines developed heavy infections with colonization of the stomodeal valve in 97% of cases. In contrast, the KO only established moderate or light infections, which were confined to the abdominal or proximal thoracic midgut, and did not reach the cardia (Fig 4E and 4F).

Quantitative analysis of promastigotes at 9 days PBM showed significant differences in the distribution of parasite morphotypes between KO, AB and parental cell lines. Morphoptype-specific transcriptional profiles have recently been

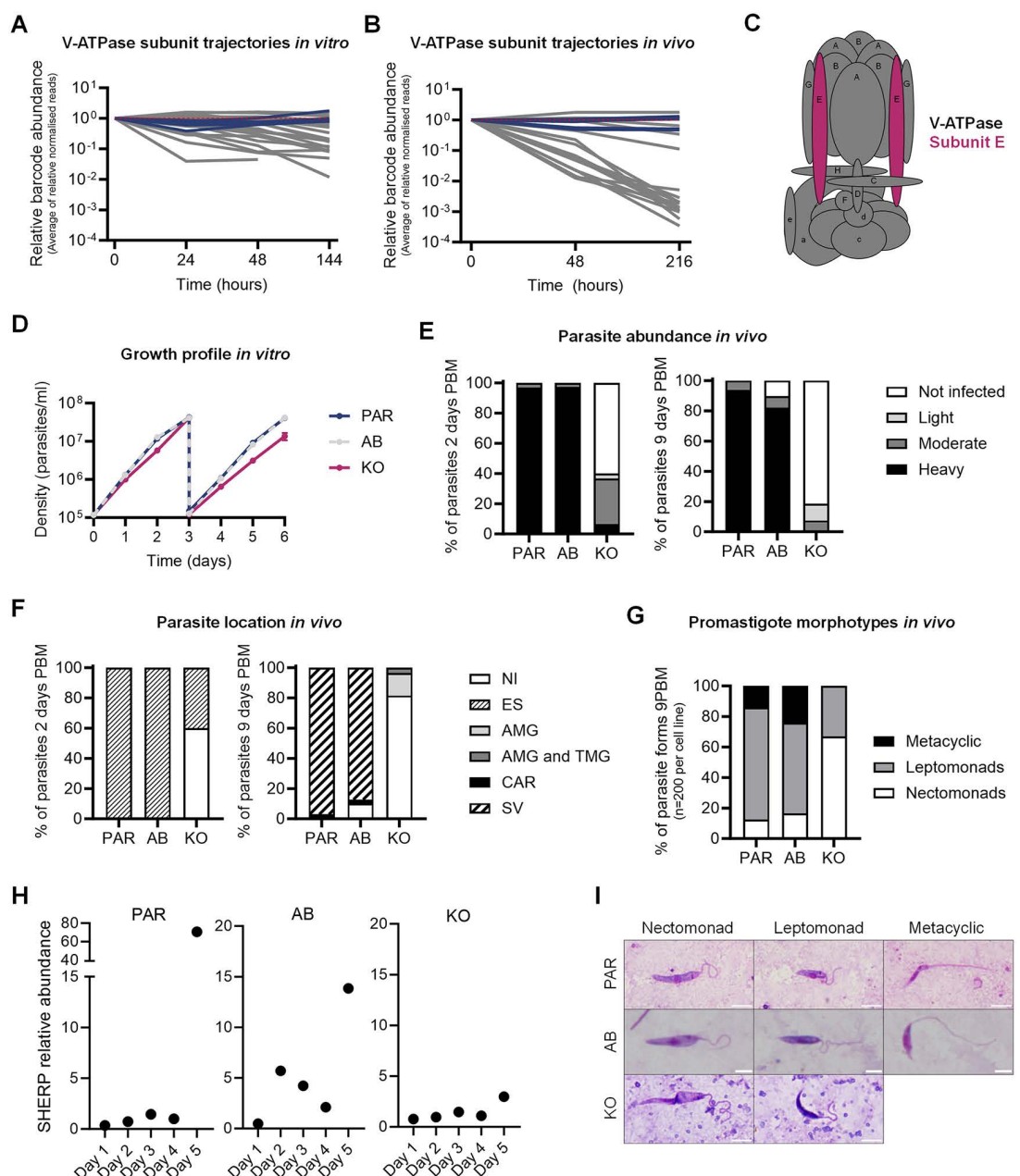

**Fig 4. V-ATPase is required for differentiation and colonisation in the sand fly vector. (A)** Trajectories of the average of normalised reads for V-ATPase deletion mutants from the promastigote masterpool (grey lines), relative to $T_0$. Dark blue lines, SBL1-5 parental controls. **(B)** Barcode trajectories for V-ATPase deletion mutants in sand flies, normalised to the start of the experiment. Colour code as in **A. (C)** Schematic of V-ATPase pump with each subunit labelled. Subunit E, for which the null mutant was individually characterised, is highlighted in magenta. **(D)** Growth of parental (PAR, dark blue), V-ATPase $V_1E$ null mutant (KO, magenta) and V-ATPase $V_1E$ add-back (AB, grey) mutant promastigotes *in vitro*. After 3 days of continuous growth, cultures were diluted back to $1 \times 10^5$ parasites/ml and monitored for an additional 3 days. **(E)** Parasite abundance in the digestive tract of dissected sand flies infected with PAR, AB or KO parasite lines, assessed at 2 days PBM (left) and 9 days PBM (right). **(F)** Location of PAR, AB or KO parasite lines at 2 days PBM (left) and 9 days PBM (right). NI, non-infected; ES, endoperitrophic space; AMG, abdominal midgut; TMG, thoracic midgut; CAR, cardia; SV, stomodeal valve. **(G)** Percentage of promastigote morphotypes of PAR, AB or KO parasite lines observed in infected gut smears after 9 days PBM. **(H)** *Sherp* transcript quantification by qPCR for PAR, AB and KO parasite lines grown *in vitro*. Each point shows the abundance of s*herp* normalised to 18S rRNA, calculated from three replicate measurements. **(I)** Representative micrographs of promastigote morphotypes found in each cell line *in vivo* at 9 days PBM. Scale bar = 5 μm.

described for *L. major* isolated from *Ph. duboscqi* guts [4]. A definitive set of molecular markers is currently lacking for most *L. mexicana* morphotypes. Morphology and location were therefore used to distinguish different promastigote developmental stages in the sand fly. Elongated cells morphometrically classified as "nectomonads" predominated in the KO, whereas leptomonads were less abundant, and metacyclic promastigotes were completely absent when compared to AB and parental lines (Fig 4G and S7 Table). Additionally, the KO exhibited a significantly longer body and flagella (16 µm, p = 0.000; 16.48 µm, p = 0.005) compared to the parental cells (10.67 µm; 14.91 µm) (S7 Table). Expression of an episomal copy of the deleted gene in the KO, significantly restored body length (11.57 µm, p = 0.000), but not flagellum length (16.11 µm, p = 0.747). To test whether metacyclic promastigotes formed *in vitro*, promastigote cultures of V-ATPase $V_1E$ KO, AB and parental cell lines were left to grow to stationary phase (S7 Fig). RNA was extracted every day for 5 days to measure the expression of the metacyclic marker *sherp* [69] by quantitative real-time PCR. The results showed an increase in the level of *sherp* transcript on day 5 in the parental and AB lines but not in the KO cell line (Fig 4H). This suggests that V-ATPAse KOs experience a delay, or block, in metacyclogenesis, consistent with the absence of metacyclics observed in sand flies.

In laboratory cultures, promastigotes in the stationary-phase may encounter metabolic stress as cultures grow dense, including waste accumulation, nutrient depletion, and cell crowding, which may cause the longer doubling times of the V-ATPase mutants (Fig 4A and 4D). In the sand fly, these challenges are intensified by competition with the host and its microbiota for limited resources and change of the external pH in the sand fly gut, which shifts from ~6 in unfed or sugar-fed insects to ~8.15 following a blood meal [70]. Blood ingestion also triggers diuresis, a process in which hematophagous insects, including sand flies, expel excess water and concentrate ingested blood, facilitating more blood uptake. This process is mediated by water absorption in the midgut and urine production by the malpighian tubules [70,71]. Our results suggest that the V-ATPase may contribute to parasite fitness under these different stresses, beyond adaptation to a low pH environment. The vacuolar proton ATPase (V-ATPase) is a conserved multi-subunit protein complex that acts as a rotary proton pump, using the energy from ATP hydrolysis to translocate protons across membranes. Their structure and biochemical activity of V-ATPase has been extensively characterized in many systems, and the primary function is the acidification of intracellular organelles. Their specific physiological functions are however varied and context-dependent, including adaptation to pH stress, regulatory functions in the endocytic pathway and autophagy [72,73]. In kinetoplastids, V-ATPases contribute to the acidification of acidocalcisomes [74,75] and the *T. brucei* V-ATPase was shown to regulate endocytosis in bloodstream forms [68,76]. The *L. mexicana* V-ATPase is concentrated near the flagellar pocket [43], compatible with a role in endocytosis and endo-/lysosomal trafficking. Moreover, autophagy [77–79] has been shown to be important for the differentiation to amastigotes and metacyclic promastigotes *in vitro* [80]. A function in autophagy could explain why metacyclogenesis was delayed or prevented in V-ATPase mutants, both *in vitro* and in the insect vector. Taken together, the mutant phenotypes *in vitro*, in the insect vector and in a mammalian host [43] identify the V-ATPase as being key to the parasite's ability to tolerate and respond to changing environments at every stage in its life cycle.

## Materials and methods

### Ethics statement

The use of BALB/c mice for sand fly breeding has been approved by Charles University Faculty of Science Expert Committee for Ensuring Good Living Conditions for Laboratory Animals and the Ministry of Education, Youth and Sports number MSMT-25062/2023–6. Mice were kept in the animal facility of Charles University in Prague in accordance with institutional guidelines and Czech legislation (Act No. 246/1992 and 359/2012 coll. on protection of animals against cruelty in present statutes at large), which complies with all relevant European Union and international guidelines for experimental animals. Investigators are certificated for experimentation with animals by the Ministry of Agriculture of the Czech Republic.

### *Leishmania* parasites

Promastigote forms of the *L. mexicana* cell line *L. mex* Cas9 T7 [51] and all generated mutants in this study were either grown in T25 cm² flasks at 27 °C or flat bottom well plates at 27 °C + 5% $CO_2$ in filter-sterilised M199 medium (Life Technologies) supplemented with 2.2 g/L $NaHCO_3$, 0.005% hemin, 40 mM 4-(2-Hydroxyethyl)piperazine-1-ethanesulfonic acid (HEPES) pH 7.4 and 10% FCS [referred in the text as "standard M199"]. 50 µg/ml Nourseothricin Sulphate and 32 µg/ml Hygromycin B were added to the medium for the maintenance of the *sp*Cas9 and T7 RNA polymerase transgenes [51].

### Phlebotomine sand flies

A laboratory colony of *Lu. longipalpis* (originating from Jacobina, Brazil) was maintained in the insectary of the Department of Parasitology, Charles University (Prague, Czechia) under standard conditions (at 26 °C, 60–70% humidity, fed on 50% sucrose solution and a 14 h light/10 h dark photoperiod) as described previously [81].

### Revised transportome of *Leishmania mexicana*

This study included the genes that were previously reported in the 'TransLeish' mutant screen [43], plus four newly identified putative membrane transporter genes: LmxM.03.0370 and LmxM.03.0390, encode for proteins of the The Acetate Uptake Transporter (AceTr) Transporter Classification Data Base (TCDB) family [50], LmxM.28.2380, belonging to the Selenoprotein P Receptor (SelP-Receptor) family and LmxM.28.2410, encoding a protein of the Multidrug/Oligosaccharidyl-lipid/Polysaccharide (MOP) Flippase TCDB family, thus updating the current size of the *L. mexicana* transportome to a total of 316 putative members.

### CRISPR-Cas9 gene knockouts

Gene deletions were done using the CRISPR-Cas9 barcoding method previously described [51]. Diagnostic PCRs for the validation of gene deletions was done as reported in [43] using ORF_Fw and ORF_Rv primers (S2 Table). In addition to targeting each gene individually, a total of 17 tandem arrays were targeted and 8 non-transporter null mutant control cell lines were produced [three independently barcoded Δ*LPG1*, two independently barcoded Δ*PF16*, and three independently barcoded Δ*IFT88* (S1 Table). Fitness screens were done with populations, without drug pressure, for which gene deletions were assessed by diagnostic PCR, without further subcloning. Exceptionally, for the deletion of the glucose transporter array (LmGT1-GT3), new primers were designed that captured the dissimilar UTR regions flanking the GT array. Drug resistant mutants were cloned by limiting dilution and ORF verification primers for validation of resulting null mutant clones were also redesigned, so that all three copies could be recognised (S6 Fig).

### Generation of V-ATPase subunit E add-back cell line

The LmxM.36.3100 gene was cloned into the pT-add plasmid via restriction digestion cloning (S1 File). Restriction sites SpeI and EcoRI were inserted into the 5' and 3' end of the gene, respectively, with the following primers:
SubE_forward_SpeI 5' TCAAGACTAGTATGAGCGAGGCACGCCAAAT 3'
SubE_reverse_EcoRI 5' AATACAGAATTCTTACAGTGGCGCCTCGGTGT 3'
The ΔLmxM.36.3100 cell line was transfected with 5 µg of the newly generated pTadd-LmxM.36.3100 plasmid as described elsewhere [78]. After approximately 15 hours following transfection, drug resistant cells were selected by addition of phleomycin at a final concentration of 25 µg/ml and cells were kept in presence of drug for the following 3 passages.

### Pooling of cells for bar-seq experiments

The barcoded mutant and parental cell lines were combined in mixed pools, adding similar numbers of each individual cell line. For the *in vitro* screen, a total of 290 individually targeted transporter mutants, 13 array transporter mutants, 5

barcoded parental lines (SBL1–5; barcodes introduced into the SSU locus [16], and 9 non-transporter knock-out mutants, of which 8 acted as controls; 3 ΔIFT88 (LmxM.27.1130), 3 ΔLPG1 (LmxM.25.0010) and 2 ΔPF16 (LmxM.20.1400) different barcoded mutants, were combined into a pool of 1x10$^5$ cells/ml (S4 Table), which was split into three aliquots for replicate measurements of *in vitro* growth.

For sand fly infections, four separate experiments were conducted using distinct pools (S4 Table, Pool membership). Pool 1 contained 75 individually targeted transporter mutant and one array mutant, Pool 2 contained 71 individual transporter mutants and three array mutants, Pool 3 contained, 77 individual transporter mutants and 5 array mutants and Pool 4 contained 76 individual transporter mutants and 4 array mutants. Each pool also contained five barcoded parental lines (SBL1–5) and three non-transporter knock-out control mutants (ΔIFT88, ΔLPG1, ΔPF16). Each of the four pools was split into three equal aliquots (replicates) in preparation for the infection of female *Lu. longipalpis* sand flies.

### Growth curves *in vitro*

For the *in vitro* growth assay, the mixed pool was split into 3 equal aliquots (replicates), which were left to grow in separate flasks for 48 h, diluted to 1 x 10$^6$ cells/ml, grown for an additional 24 h, diluted to 1x10$^5$ cells/ml and grown for an additional 72 h (total 144 hours) at 27 °C + 5% $CO_2$ in standard M199. For the growth curves of individual cell lines, log phase promastigotes (between 2 and 4 x 10$^6$ parasites/ml) of V-ATPase Subunit E (LmxM.36.3100) knockout, add-back (AB) and *L. mex* Cas9 T7 (C9T7) cell lines were seeded in standard M199 at a density of 1 x 10$^5$ cells/ml. Parasites were left to grow at 27 °C for 3 consecutive days and on day 3, were diluted back to 1 x 10$^5$ cells/ml density and left to grow at 27 °C. Growth was assessed by counting the cells every 24 hours with a CASY cell counter (Cambridge Bioscience) using a 60 µm capillary and measurement range set between 2 and 15 µm. For each condition measurements from three replicate flasks were recorded.

### Sand fly infections

For infections with pooled barcoded mutant populations, each pool was seeded at a density of 2 x 10$^6$ parasites/ml and grown for 24 h at 26 °C in standard M199 with 250 µg/ml of Amikacin (Amikin). On the day of infection, a total of either 3 x 10$^7$ (Pools 1–3), or 1.8 x 10$^7$ (Pool 4) logarithmic growing parasites were washed three times with sterile 0.9% NaCl saline solution (Braun) and then resuspended in 200 µl of saline, which were then mixed with 2.8 ml of defibrinated ram blood (LabMedia), previously heat inactivated at 56 °C for 35 minutes. For each separate pool, three groups of 120–180 female sand flies, 4–5 days old, were allowed to feed on the parasite-blood mixture, through a skin membrane from a 1-day old chick, as previously described [34]. Fully engorged females were separated and maintained at 26 °C with free access to 40% sucrose solution. Infected sand flies were dissected at days 2 (48 h) and 9 (216 h) post blood-meal (PBM) (S8 Table). At day 2 PBM, a total of 1–5 female sand fly guts were checked to qualitatively assess the progress, localisation and intensity of infection by light microscopy. Parasite abundance was graded into three qualitative categories: negative, light (<100 parasites/gut), moderate (100–1000 parasites/gut) and heavy (>1000 parasites/gut), as described elsewhere [34]. For infections with individual promastigote cell lines; female sand flies (5–9 days old) were infected by feeding through a chick-skin membrane parasites from log-phase cultures (3–4 day cultures), washed twice in sterile saline solution and resuspended in heat-inactivated ram blood at concentration of 1 x 10$^6$ promastigotes/ml. Engorged sand flies were maintained as described above. Female flies were dissected at days 2 and 9 PBM and the abundance and localisation of *Leishmania* promastigotes in the sand fly digestive tract was examined as described above. Experiments were performed in duplicates.

### Sampling of DNA for sequencing

For the promastigote *in vitro* cultures, gDNA was extracted at 0 h, 24 h, 48 h and 144 h of growth from approximately 1x10$^7$ cells from each replicate culture, using the Wizard SV Genomic DNA Purification System (Promega) according to the manufacturer's instructions, eluting in 40 µl of bi-distilled water (Ambion). For the *in vivo* experiments, gDNA was

extracted from the pool after mixing in standard M199 and from the parasite-blood mixture used for the infection (time point 0). For extraction of genomic DNA from whole infected sand flies, a total of 17–82 specimens were collected from each batch at 2- and 9-days post blood meal (PBM) (S5 Table). For both cells and tissues, the High Pure PCR Template Preparation Kit (Roche) was used and all samples were eluted in 100 µl of VWR Life Sciences PCR grade water as previously described [16].

### Bar-seq library preparation and sequencing

The preparation of bar-seq amplicon libraries was done as previously reported [43], with minor changes. For the initial bar-seq amplicon PCR, 100 ng of gDNA isolated from promastigote cultures, 250 ng of gDNA isolated from *Leishmania*-blood meal mix and 500 ng gDNA isolated from whole *Leishmania*-infected female sand flies, were used. To account for the excess of host gDNA present in blood and sand fly derived samples, the number of cycles for the same PCR was increased from 31 for *Leishmania* culture derived samples to 35 for blood and sand fly samples. Raw sequencing files (fastq) for all samples generated during this study were deposited in the European Nucleotide Archive (ENA) study accession PRJEB90861 (ERP173867).

### Quantification of barcoded cell line fitness

For each sample, we grouped and counted raw sequencing reads containing barcode sequences, normalised this to the sample total reads and calculated the change in the proportion of reads of each cell line in each sample compared to the zero time point, as previously reported [44]. The majority of cell lines in a pool do not exhibit a fitness change, therefore we identified the mode (peak) of the distribution of changes in proportion as the reference to compare to in each replicate. For each time point and cell line we assigned fitness scores by dividing the median of the cell line's change in proportion over all replicates with the median over all modes. A fitness score above one indicates that proportion of barcodes from a particular cell line have increased faster relative to the bulk of the pooled cell lines, corresponding to faster growth and/or better survival than the bulk of the pooled cell lines from the start of the assay up to that time point. A fitness score below one indicates the inverse. P-values were calculated using a paired t-test of the log-transformed cell line changes in proportions against the corresponding reference values, testing the null hypothesis that the cell line change in proportion from all replicates of a particular cell line in a given time point cannot be distinguished from the change of the bulk of the pooled cell lines in all replicates of the same time point. Cell lines were labelled as having a strong fitness phenotype in a given time point if their p-value was below 0.05 and their fitness score was either below 0.5 (deleterious phenotype) or above 2 (beneficial phenotype).

### Population bottleneck assessment

To determine the appropriate number of sand flies for infection to avoid mutant lines within a pool going extinct by chance due to population bottlenecks, we first needed to determine the number of parasites that establish an infection in a sand fly. To do so, we first assessed how evenly barcodes were represented within each replicate of each pool by fitting a symmetrical Dirichlet distribution to the barcode proportions using maximum likelihood estimation. The single resulting dimensionless parameter determines the pool evenness: a small value indicates that very few cell lines dominate the pool while a very large value corresponds to all barcodes being equally represented. Next, we treat the experiment as a Pólya urn model, we identify parasites in a hypothetical pool sample (using the same evenness as measured from our actual pools) as coloured balls in an urn, with the colour representing the M different barcodes from mutant lines in the pool. We then model the establishment of infection and detection of the barcode by drawing N balls from this urn (corresponding to detecting the barcode from that parasite cell) and count the number of different colours present C (corresponding to the number of different mutant lines detected). The experimentally observed fraction of surviving mutant lines is S = C/N. The

fraction of surviving mutant lines can also be modelled as a function of the population bottleneck N, the number of cell lines M and the pool evenness α is given by

$$S_\alpha(M, N) = \sum_{i=1}^{M} (1 - [1 - p_i]^N),$$

with the abundance of individual mutant lines $p_i$ distributed according to a symmetric Dirichlet distribution with the concentration parameter α. Lengthy but straightforward calculation yields that the expectation value of the survived fraction is then given as

$$E\left[S_\alpha(M, N)\right] = 1 - \exp\left(\frac{\Gamma(M\alpha)\Gamma\left(N + [M-1]\alpha\right)}{\Gamma\left([M-1]\alpha\right)\Gamma(N + M\alpha)}\right).$$

Knowing the experimentally determined survived fraction S, the number of cell lines in the pool M and the pool evenness α, we can now determine the population bottleneck size N by numerically inverting the above relationship. Therefore, we can estimate the number of cells that, on average, successfully establish an infection as N/K where K is the number of sand flies used in an experiment, because each sand fly can be treated as independent from all others (Fig 3B).

This calculation uses the measured pool evenness from time point zero and treats our hypothetical pool cell lines to have equal fitness and hence equal chance of survival. However, this is an approximation as the actual fraction of surviving mutants also depends on the intrinsic fitness of the individual mutant lines, which will tend to increase extinction events as deletion mutations tend to be detrimental. As a result, this procedure underestimates of the population bottleneck size N and therefore overestimates the number of sand flies required.

### *In vivo* parasite morphometry

Midgut smears of infected sand flies were fixed with methanol, stained with Giemsa, examined by light microscopy (Olympus BX51) with an oil immersion objective and photographed (Olympus DP72). Body length, flagellar length and body width of 200 randomly selected promastigotes were measured on day 9 PBM using Fiji [34]. Four morphological forms were distinguished, based on criteria previously described [10,28]. Briefly; elongated nectomonads (EN), body length ⩾14 µm; leptomonads (LE) body length <14 m and flagellar length ≤2 times body length and metacyclic promastigotes (MP), flagellar length >2 times body length and body length < 14 µm. Haptomonads were not distinguished in this study as they are often remaining attached to the gut and can be underrepresented on gut smears. Differences in proportion of morphological forms and measurements were compared by a Tukey's HSD (honestly significant difference) test, using the software SPSS version 27.

### Quantification of *sherp* expression by qPCR

Cultures of V-ATPase $V_1E$ KO, AB and parental cell lines were seeded at a density of 1 x $10^5$ cells/ml on day 0 and grown at 27 °C/ 5% $CO_2$ in standard M199, with five replicate flasks for each line. Starting from day 1 of growth, total RNA was extracted from $1 \times 10^8$ cells every day for 5 consecutive days, using PureLink RNA Mini Kit (Ambion by Life Technologies, Invitrogen) according to the manufacturer's protocol, with the exception of the homogenization step. To clear the genomic DNA in the sample, 10 µg of purified RNA was subjected to DNase I treatment (New England Biolabs), according to the manufacturer's protocol. Afterwards, RNA cleanup was performed with the Monarch RNA cleanup kit (New England Biolabs), according to the manufacturer's protocol. To ensure a complete cleanup of genomic DNA, the DNase I treatment and subsequent RNA cleanup were performed twice. cDNA was synthesized from 200 ng

RNA samples using the iScript cDNA synthesis kit (Biorad) according to the manufacturer's instructions. The qPCR was done with the SsoAdvanced Universal SYBR Green Supermix (Biorad), according to the manufacturer's protocol. 1 ng cDNA sample was used for each qPCR reaction. The *sherp* primer sequences were taken from [69]: Forward: 5'- CTGCAGCCTTGTTGCTCA-3'; Reverse: 5'- ACCAGGAGACAAGGGACCA-3'. The 18S rRNA was used as a control, with primer sequences taken from [82]: Forward: 5'- CTGCAGCCTTGTTGCTCA-3'; Reverse: 5'- ACCAGGAGACAAGG GACCA-3'. All cDNAs were run in triplicates with each primer pair, in a qPCR-compatible 96 well plate, using Biorad C1000 touch thermal cycler. For the calculation of relative *sherp* transcript abundance, the quantification cycles (Cqs) of each reaction with *sherp* or 18S rRNA was averaged. Then, the average Cq of 18S rRNA was subtracted from *sherp* Cqs for the corresponding sample ($\Delta$Cq), for normalization. Relative abundance was calculated by the formula $2^{-\Delta Cq}$; this value was multiplied by 1000 to generate the plots.

## Supporting information

**S1 Fig. Consolidated summary of gene deletion results for the transportome of *Leishmania mexicana*.** (A) Top, pie-charts showing the numbers of successful gene deletions (cyan) and non-successful deletion attempts (magenta), across two independent screens (44 and this study). Bottom, break-down of non-successful deletions attempts into two sub-categories: (i) Double drug-resistant populations where ORF is still detected (or PCR inconclusive) (yellow); (ii) Attempts where no drug resistant populations were ever recovered, or populations where resistant cells could only be recovered with single drug selection and ORF was still detected (dark pink). (B) Summary of gene deletion results separated into TCDB families (S3 Table); colours as for A.
(PDF)

**S2 Fig. Diagnostic PCR gel electrophoresis.** Results of diagnostic PCRs for genotypic validation of all new mutant cell lines reported in this study.
(PDF)

**S3 Fig. Overview of targeted genes organised in tandem arrays.** (A) Pie-charts summarising gene deletion results separated into TCDB families (left) and in total (right) (S3 Table). (B) Overview of all genotypes for single genes and whole targeted arrays, coloured in four main categories (as in S1 Fig).
(PDF)

**S4 Fig. Trajectories of the average of normalised reads *in vitro* and *in vivo*.** (A) Plot legends. (B-F) Trajectories of the average of reads for all mutant barcodes in all conditions studied in this report, normalized to total reads for the experiment, reflecting the range of relative barcode proportions for the different mutants in each pool.
(PDF)

**S5 Fig. Trajectories of the average of normalised reads *in vitro* and *in vivo*, relative to time zero.** (A) Plot legends. (B-F) Trajectories of the average of barcode reads for all mutants in all conditions studied in this report, normalized to total reads for the experiment and relative to time-point "0 hours" ($T_0$).
(PDF)

**S6 Fig. Generation, validation and characterisation of the glucose transporter 1–3 array null mutant.** (A) Diagnostic PCR gel electrophoresis of GT gene array null mutant clones. Lane 1, clone B4 gDNA + GT_array_ORF primers; lane 2, clone B7 gDNA and GT_array_ORF primers; lane 3, clone B4 gDNA and PFR2 primers (positive control for gDNA); lane 4, clone B7 gDNA and PFR2 primers; lane 5, parental gDNA and GT_array_ORF primers; lane 6, parental gDNA and PFR2 primers; lanes 7–9, gDNA of clones B4, B7 and parental without addition of primers. (B) Lane 1, clone H1 and GT_array_ORF primers; lane 2, clone H1 and PFR2 primers; lane 3, parental gDNA and GT_array_ORF

primers; lane 4, parental gDNA and PFR2 primers; lanes 5 and 6, negative controls (no gDNA). DNA ladder: GeneRuler 100 bp DNA ladder by Thermo Scientific (REF: SM0243). (C) Locations of primer binding sites for diagnostic PCR. The forward primer (GT_array_ORF_FW) is highlighted with grey circles and the reverse primer (GT_array_ORF_RV) in blue circles. Note that both FW and RV primers recognise all glucose transporter open reading frames (ORFs) in the array. (D) Growth profile of GT KO clones over 72 hours. Black, Parental cell line; Yellow, LmGT array KO clone B4; Turquoise, LmGT array KO clone B7. Data points show the average of three measurements. (E) Doubling times of cells, calculated from D.

(PDF)

**S7 Fig. Continuous growth curve of *Leishmania* cell lines.** Growth of parental (PAR, dark blue), V-ATPase V1E null mutant (KO, magenta) and V-ATPase V1E add-back (AB, grey) mutant promastigotes in vitro for 5 days of continuous growth. At each time point samples were collected for qPCR analysis (see Fig 4H).

(PDF)

**S1 File. Map of the pTadd-LmxM.36.3100 after restriction digestion cloning.** FASTA DNA sequence of pTadd-LmxM.36.3100 plasmid used to restore LmxM.36.3100 activity in the ΔLmxM.36.3100 deletion mutant.

(FA)

**S1 Table. Revised TransLeish DB.** Information about single and array genes targeted in this study.

(XLSX)

**S2 Table. Barcodes and primers.** Barcodes and primer sequences used for the generation of mutant cell lines and diagnostic PCRs.

(XLSX)

**S3 Table. Diagnostic PCR results.** Results of diagnostic PCRs for knockout validations for all mutant cell lines as well as gel electrophoresis schemes (from S1 Fig).

(XLSX)

**S4 Table. Pool membership.** List of barcoded mutant cell lines, barcode sequences and ids included in each analysed pool *in vitro* and *in vivo*.

(XLSX)

**S5 Table. Fitness scores and raw read counts. Fitness scores, p-values from t-test, raw read counts and summary of reads for each timepoint from promastigotes grown *in vitro* (0, 24, 48 and 144 hours) and sand fly infections (0, 48 and 216 hours).**

(XLSX)

**S6 Table. Growth curve raw data and doubling time calculations.** Data from densities recorded for the promastigote *in vitro* cultures, corresponding doubling times, as well as doubling times.

(XLSX)

**S7 Table. Cell size measurements from promastigotes *in vivo*.** Measurements of body length, width and flagellar length of Parental, V-ATPase $V_1E$ null and V-ATPase $V_1E$ add-back promastigote mutant cell lines isolated at 9 days PBM.

(XLSX)

**S8 Table. Records of the number of sand flies collected in each sub-pool.** Information about the number of female sand flies infected with sub-pools P1-P4 and masterpool at 2 and 9 days PBM for bar-seq analysis.

(XLSX)

## Acknowledgments

We like to thank Pamela Nicholson and Daniela Steiner (Next Generation Sequencing Platform, University of Bern) for help with Illumina sequencing, Vít Dvořák for maintaining the colony of *Lu. longipalpis*, Kristýna Srstková for technical support during sand fly experiments and all past and current members of the Gluenz and Volf labs for helpful discussions.

## Author contributions

**Conceptualization:** Eva Gluenz, Andreia Albuquerque-Wendt.

**Data curation:** Ulrich Dobramysl, Eva Gluenz, Andreia Albuquerque-Wendt.

**Formal analysis:** Jovana Sádlová, Ulrich Dobramysl, Eva Gluenz, Andreia Albuquerque-Wendt.

**Funding acquisition:** Richard J. Wheeler, Petr Volf, Eva Gluenz, Andreia Albuquerque-Wendt.

**Investigation:** Jovana Sádlová, Ulrich Dobramysl, Barbora Bečvářová, Tomáš Bečvář, Çağla Alagöz, Sandro Möri, Andreia Albuquerque-Wendt.

**Methodology:** Jovana Sádlová, Ulrich Dobramysl, Richard J. Wheeler, Petr Volf, Eva Gluenz, Andreia Albuquerque-Wendt.

**Project administration:** Petr Volf, Eva Gluenz.

**Resources:** Petr Volf, Eva Gluenz.

**Software:** Ulrich Dobramysl, Richard J. Wheeler.

**Supervision:** Jovana Sádlová, Richard J. Wheeler, Petr Volf, Eva Gluenz, Andreia Albuquerque-Wendt.

**Visualization:** Andreia Albuquerque-Wendt.

**Writing – original draft:** Andreia Albuquerque-Wendt.

**Writing – review & editing:** Jovana Sádlová, Ulrich Dobramysl, Barbora Bečvářová, Tomáš Bečvář, Çağla Alagöz, Sandro Möri, Richard J. Wheeler, Eva Gluenz, Andreia Albuquerque-Wendt.

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
