## [Decision Letter · Decision Letter 0]

11 Sep 2025

Identification of transporters essential for survival of Leishmania promastigotes in the digestive tract of sand flies

PLOS Pathogens

Dear Dr. Albuquerque-Wendt,

Thank you for submitting your manuscript to PLOS Pathogens. After careful consideration, we feel that it has merit but does not fully meet PLOS Pathogens's publication criteria as it currently stands. Therefore, we invite you to submit a revised version of the manuscript that addresses the points raised during the review process.

Please submit your revised manuscript within 60 days Nov 10 2025 11:59PM. If you will need more time than this to complete your revisions, please reply to this message or contact the journal office at plospathogens@plos.org. Please include the following items when submitting your revised manuscript:

We look forward to receiving your revised manuscript.

Kind regards,

Álvaro Acosta-Serrano

Academic Editor

PLOS Pathogens

Jeffrey Dvorin

Section Editor

PLOS Pathogens

Sumita Bhaduri-McIntosh

Editor-in-Chief

PLOS Pathogens

orcid.org/0000-0003-2946-9497

Michael Malim

PLOS Pathogens

orcid.org/0000-0002-7699-2064

**Additional Editor Comments:**

Reviewer #1:

Reviewer #2:

Reviewer #3:

**Journal Requirements:**

- ® on pages: 21, and 22.

Potential Copyright Issues:

i) Figures 2A, and 3A. Please confirm whether you drew the images / clip-art within the figure panels by hand. If you did not draw the images, please provide (a) a link to the source of the images or icons and their license / terms of use; or (b) written permission from the copyright holder to publish the images or icons under our CC BY 4.0 license. Alternatively, you may replace the images with open source alternatives. See these open source resources you may use to replace images / clip-art:

2) If any authors received a salary from any of your funders, please state which authors and which funders..

**Reviewers' Comments:**

Reviewer's Responses to Questions

**Part I - Summary**

Reviewer #1: The results are clearly presented and interpreted and the manuscript is similarly well written.

This is the sandfly extension of a large study into the role of transporters in Leishmania. The global approach to generate and barcode CRISPR/Cas9 gene edited cell lines is powerful in that it allowed 300+ lines to be compared, simultaneously, in the same sandflies or culture flask. This removed much of the experimental variation that would occur if they were worked on individually. The methods for culture and fly infection seem appropriate as do the methods of comparing the overall growth fitness and I appreciate that there needs a high throughput approach to analyze and interpret so many lines. I particularly liked the inclusion of reference mutant lines, based on their well characterized role for parasite colonization, to assess the robustness of the approach, particularly between the pools of mutant.

However, what I don’t get a sense of is the how these gene disruptions, even the one that was singled out, the V-ATPase Subunit E, work in the vector.

This is partly hampered by the lack of a sampling point shortly after blood defecation. This is an important bottleneck in the Leishmania-sandfly interaction where those parasites that survive have the opportunity to colonize the rest of the sandfly and complete their development. As such, this would be a good comparator for both the day 2 and day 9 infections. I am not expecting the authors to revisit their experiments but I think this needs to be discussed in the context of their study. Despite this the authors manage to identify many new transporters that are beneficial or detrimental to sandfly infection.

Reviewer #2: Sadlova and colleagues present the results of a bar-seq knock-out screen of Leishmania mexicana transporter genes and tandem gene arrays. The authors perform in vitro passage of promastigote cultures to assess the competitive fitness of the lines and then use these lines to infect Lutzomyia longipalpis sand flies. The fitness scores from this experiment identify 34 lines that are exclusively compromised in sand fly infections. The Vacuolar H+ ATPAse (V-ATPase) is singled out for validation and was identified as one of the transporters essential for parasite persistence and differentiation in the fly.

Overall the work is technically robust, and yields novel insight into transporters that are required for normal relative fitness of promastigote forms in the sand fly. However, this is tempered by the work being incremental in nature and lacking mechanistic insight into the reason why V-ATPase is required in the sand fly but not in vitro. As a previous iteration of this library (by the same research team) has already been screened in bar-seq lifecycle profiling (albeit without the sand fly stage) the inclusion of a sand fly model is welcome, but not fully exploited here to yield substantial interest to the broader community of researchers studying host-pathogen interactions.

Strengths

Application of the sand fly model to the transportome KO library - novel insight into which transporters are important in this system.

Attempts to delete tandem gene arrays of various transporters.

Reproducibility of previous KO screens & phenotypes could be assessed.

Addback study to confirm V-ATPAse phenotype.

Weaknesses

Many of the KO phenotypes in vitro are already determined from previous work.

The 34 genes giving essential/important fitness defects in the sand fly are not explored in detail when these are the key insights arising from the study.

Addback of V-ATPase V1E could be used as a platform for functional genetics to explore the complex in detail but isn't.

Novelty

The researchers previously generated a barcoded library of transporter KO strains (TransLeish Project) which were profiled in vitro and in macrophage infections - therefore the in vitro aspect of the current manuscript is not really that novel. The reason for generation of a new library of strains is not explained in the current manuscript - it would be helpful to elaborate on this. It would also be useful to show the correlation between the results of the TransLeish screen and this screen, even at the level of ability to make a knockout. It seems that the authors were able to generate 46 extra mutants in this attempt compared to the last one, can they speculate on why that may be.

The application of this KO library to Sand Fly infections provides the element of novelty in the manuscript and the 34 genes associated with fitness cost exclusively in the sand fly can now provide insight into the unique aspects of this interaction, but this is left unexplored.

V-ATPase is defined as important in vivo but not in in vitro culture, an addback is performed of the V1E subunit KO mutant to validate this is a specific phenotype. However, there is no exploration of the protein complex function, for example using point mutations in V1E to explore the subunit molecular function (which the authors don't mention anyway). Most of the other descriptive characterisation of this complex was performed in a previous paper by the same team (localisation etc). The individual stresses which compromise the V1E are not explored with in vitro models using these mutants.

Reviewer #3: The study by Sadlova et al. investigated which transporter proteins are essential for Leishmania promastigotes. Using a library of over 300 Leishmania mutants, the researchers tested their ability to grow in laboratory conditions and to develop in the digestive tract of a sand fly. They reported that 34 transporter genes being of relevance for parasite survival inside the sand fly, and that the V-ATPase is especially important. The study is ambitious and generally well-executed with comprehensive molecular assessment.

The study lacks, however, important biological contextualization and some technical points must be clarified.

**Part II – Major Issues: Key Experiments Required for Acceptance**

Reviewer #1: An interesting result of this paper was the enlarged phenotype of the nectomonad stage and diminished metacyclogenesis of the V-ATPase Subunit E knockout line but these were not worked on beyond observation. Perhaps this could be extended by looking for the effect of this mutation on metacyclogenesis in vitro; the ultrastructure (particularly, the volume or surface area of the mitochondrion) of the nectomonad cells; or see if this result can be replicated for nectomonads in vitro by subjecting them to stressors experienced in sandflies, such as pH or osmolarity changes, exposure to blood digestion products or lack of nutrients? Collectively, this would help begin to understand the role of this transporter for Leishmania.

Reviewer #2: The authors have presented some solid descriptive biology, but I don't believe there is enough new, transformative, or mechanistic information to justify publishing in PLoS Pathogens; suggest manuscript is submitted elsewhere, for example PloS NTDs.

Specifically, several Leishmania bar-seq screens have been published in PLoS Pathogens which include profiling in vitro and in murine infections, but have also been followed up with detailed biochemical assessment of hit proteins , inducible deletion of essential proteins, and/or immunoprecipitation of novel protein interactions (Burge et al 2020, Damianou 2020). The validation of a single KO strain (V1E) by addback restoration, without any further functional genetics, biochemical or cellular exploration does not add much to the existing knowledge on this protein complex. If another, novel transporter had been chosen and characterised the manuscript would have been greatly strengthened.

Perhaps adding experiments to define the individual stress or combination of stresses that compromise the V1E null mutant could be added to flesh out the biological insight of the work. Or if the authors have characterisation data of a novel transporter identified to be important in vivo it could be included to make it align better with PLoS Pathogens criteria for publication.

Reviewer #3: 1) Line129: The authors state that promastigotes undergo terminal differentiation. This terminology should be avoided as it doesn’t fit Leishmania life cycle. Inside the sandfly it has been shown that metacyclics dedifferentiate and regain ability to multiply after complete morphological reshaping. Beyond that, transmitted promastigotes will not remain as is. They will transform into amastigotes. “Terminal differentiation” imply permanently exiting the cell cycle and losing its ability to divide.

2) Figure 2: The variation in nutrient stress and other biotic and abiotic stochastic events is always present in the promastigote cycle. It will be very informative for this work to see a trasnportome actually working on a variable environment. Have you restarted the culture from 144hr time point? Also, 199 media, although widely used for Leishmania, is a media developed for fibroblasts. If using insect cell media, 30% or less of essentiality is still sustained?

3) How does the gain/loss of fitness of the mutant lines behave under multiple blood meals? Other authors have shown how important this constant change in the environment is crucial for vector competence. Beyond that, it’s a certain event to happen in the sand fly biology and the course of promastigotes cycle. Authors should evaluate its effect on mutants’ fitness.

**Part III – Minor Issues: Editorial and Data Presentation Modifications**

Reviewer #1: Could the authors clarify from the methods that ‘3 to 9 guts’ were sampled at day 2 PBM? If so, do they think this is large enough to accurately determine the proportion of infected flies, the location of the infection and infection intensity? Was this the same for day 9 PBM, as this information is missing from the methods or the figure legends?

Reviewer #2: The number of sandflies used is a bit unclear - what was the number of fed flies used for DNA extraction per each sub-pool replicate of the bar-seq screen? Can this be detailed in the figure legend? Did the authors perform any power calculations to determine the minimum effect size that could be reliably determined by this assay type and the number of flies/replicates used?

For the in vitro growth were the selective antibiotics applied to the culture? If they were not does this influence the loss of barcodes from "refractory to deletion" strains due to the mosaic aneuploidy of Leishmania leading to loss of heterozygosity ? Thus delinking fitness of the barcoded strain to the amount of barcode?

The manuscipt discusses "arrayed CRISPR/Cas9 mutant libraries" (the technique) and also "array mutants" (when talking about tandem gene arrays), and when the latter were introduced there were a couple of confusing instances; I would suggest that when referring to the tandem gene arrays this is explicit in every instance.

Is CRISPR a relevant term given that the authors are not using clustered regularly interspersed palindromic repeats in their system? Would Cas9-mediated homology-directed repair be more accurate?

Reviewer #3: 4) Figure 1 does not add much info to the manuscript. Part A is sufficient to tell the message and part B is too noisy for a main figure. I suggest you keep only A or move all to supplementary.

5) Figure 4G: Manuscript will benefit from display of images from the parasites.

6) Lines 384-396: This is good discussion. However, the authors will benefit by adding to this the fact that in vitro more than the correctly pointed is “not happening”. Many parasite stages just don’t appear in culture. Considering transporters, lacking in culture stages that will be present inside the vector may just lead to unrealistic loss/gain of fitness.

PLOS authors have the option to publish the peer review history of their article (what does this mean? ). If published, this will include your full peer review and any attached files.

**Do you want your identity to be public for this peer review?** For information about this choice, including consent withdrawal, please see our Privacy Policy .

Reviewer #1: No

Reviewer #2: No

Reviewer #3: No

**Figure resubmission:**

**Reproducibility:**



---

## [Decision Letter · Decision Letter 1]

28 Feb 2026

Dear Albuquerque-Wendt,

We are pleased to inform you that your manuscript 'Identification of transporters essential for survival of Leishmania promastigotes in the digestive tract of sand flies' has been provisionally accepted for publication in PLOS Pathogens.

**Note from the editor** : The revised version has been strengthened by additional data and improved biological context, as noted by the reviewers. While a full mechanistic dissection of V-ATPase function in *Leishmania* biology within the sand fly remains to be determined (a limitation acknowledged by the authors) and can be addressed in a future study, this manuscript represents a pioneering and technically rigorous analysis of molecular interactions between *Leishmania* and its vector. It also provides a valuable framework for bar-seq–based functional screens in this system. As such, it offers an important and timely contribution to the field.

Best regards,

Álvaro Acosta-Serrano

Academic Editor

PLOS Pathogens

Jeffrey Dvorin

Section Editor

PLOS Pathogens

Sumita Bhaduri-McIntosh

Editor-in-Chief

PLOS Pathogens

orcid.org/0000-0003-2946-9497

Michael Malim

Editor-in-Chief

PLOS Pathogens

orcid.org/0000-0002-7699-2064

Reviewer Comments (if any, and for reference):

Reviewer's Responses to Questions

**Part I - Summary**

Reviewer #2: The authors have made efforts to address all the specific reviewers comments with a careful and detailed response. This improved the manuscript in terms of the clarity of the technical details. I note that all reviewers raised concerns about the amount of biological characterisation of the V1E mutant, and nothing was really added to expand on this beyond narrowing down that the block in differentiation occurs in a pre-metacyclic stage.

The estimation of bottlenecks in the sand fly is a useful addition, and provides meaningful information to support these kind of experiments in the future.

My overall view on the manuscript hasn't really changed. This is a high-quality, technical evolution that will provide useful information on how to conduct the bar-seq screens in sandflies, and a reference point for those interested in Leishmania transporter biology; but none of the revisions really address the lack of a mechanistic characterisation of the Vatpase protein (or another interesting target emerging from the screen).

As my critique on the article is more editorial in its context than scientific, I will defer to Editors to make the final decision on acceptance.

Reviewer #3: The revised manuscript is improved and the authors have responded constructively to the key conceptual concerns raised during review. The revision adds useful biological context for interpreting in vitro fitness relative to sand fly infection, and data presentation has been strengthened (e.g., relocation of the former Figure 1 content to the Supplement and inclusion of parasite images in the revised Figure 4), improving clarity and transparency. While some suggestions were not performed, the limitations are acknowledged.

**Part II – Major Issues: Key Experiments Required for Acceptance**

Reviewer #2: (No Response)

Reviewer #3: No additional key experiments are required.

**Part III – Minor Issues: Editorial and Data Presentation Modifications**

Reviewer #2: (No Response)

Reviewer #3: No additional issues.

PLOS authors have the option to publish the peer review history of their article (what does this mean? ). If published, this will include your full peer review and any attached files.

**Do you want your identity to be public for this peer review?** For information about this choice, including consent withdrawal, please see our Privacy Policy .

Reviewer #2: No

Reviewer #3: No

---

## [Editor Report · Acceptance letter]

Dear Albuquerque-Wendt,

We are delighted to inform you that your manuscript, "Identification of transporters essential for survival of Leishmania promastigotes in the digestive tract of sand flies," has been formally accepted for publication in PLOS Pathogens.

Best regards,

Sumita Bhaduri-McIntosh

Editor-in-Chief

PLOS Pathogens

orcid.org/0000-0003-2946-9497

Michael Malim

Editor-in-Chief

PLOS Pathogens

orcid.org/0000-0002-7699-2064